# Characterization of increased mucus production of HT29-MTX-E12 cells grown under Semi-Wet interface with Mechanical Stimulation

**Janneke Elzinga**[1☯]*, **Benthe van der Lugt**[2☯], **Clara Belzer**[1‡], **Wilma T. Steegenga**[2‡]

**1** Laboratory of Microbiology, Wageningen University and Research, Wageningen, The Netherlands,
**2** Division of Human Nutrition and Health, Wageningen University and Research, Wageningen, The Netherlands

☯ These authors contributed equally to this work.
‡ These authors also contributed equally to this work.
* janneke.elzinga@wur.nl

**Data Availability Statement:** All microarray files are available from the NCBI GEO database (accession number GSE173729).

## Abstract

The intestinal mucus layer plays a crucial role in human health. To study intestinal mucus function and structure *in vitro*, the mucus-producing intestinal cell line HT29-MTX-E12 has been commonly used. However, this cell line produces only low amounts of the intestine-specific MUC2. It has been shown previously that HT29-MTX-E12 cells cultured under Semi-Wet interface with Mechanical Stimulation (SWMS) produced higher amounts of MUC2, concomitant with a thicker mucus layer, compared to cells cultured conventionally. However, it remains unknown which underlying pathways are involved. Therefore, we aimed to further explore the cellular processes underlying the increased MUC2 production by HT29-MTX-E12 cells grown under SWMS conditions. Cells grown on Transwell membranes for 14 days under static and SWMS conditions (after cell seeding and attachment) were subjected to transcriptome analysis to investigate underlying molecular pathways at gene expression level. Caco-2 and LS174T cell lines were included as references. We characterized how SWMS conditions affected HT29-MTX-E12 cells in terms of epithelial barrier integrity, by measuring transepithelial electrical resistance, and cell metabolism, by monitoring pH and lactate production per molecule glucose of the conditioned medium. We confirmed higher MUC2 production under SWMS conditions at gene and protein level and demonstrated that this culturing method primarily stimulated cell growth. In addition, we also found evidence for a more aerobic cell metabolism under SWMS, as shown previously for similar models. In summary, we suggest different mechanisms by which MUC2 production is enhanced under SWMS and propose potential applications of this model in future studies.

**Funding:** JE has been funded by a Building Blocks of Life grant from the Netherlands Organization for Scientific Research (NWO), grant no. 737.016.003. BvdL has been funded by the Nutricia Research Foundation, grant no. 2018-25 and the NWO Graduate Programme on Food Structure, Digestion and Health, grant no. 022.006.009. The funders had no role in study design, data collection and analysis, decision to publish, or preparation of the manuscript.

**Competing interests:** The authors have declared that no competing interests exist.

## Introduction

The surface of the human gastro-intestinal (GI) tract is covered by a layer of mucus, protecting the host from pathogens, harmful chemical or biological substances and physical damage [1–4]. Defects in this mucus layer have been implicated in several intestinal pathologies. For instance, *Muc2* knock-out mice were shown to develop spontaneous colitis [5] and colorectal cancer [6], emphasizing the important protective role of mucus. Along the GI tract, the mucus layer is thickest in the colon [7], where the number of intestinal bacteria is also highest [8]. Interestingly, the mucus layer does not only protect the underlying epithelium from these high bacterial numbers, but also provides a binding site and nutrition-rich niche for residing intestinal bacteria, such as mucin-degrading bacterial species which liberate short-chain fatty acids for cross-feeding bacteria and the host [9–11]. As a nutrient source, mucin has been shown to be a major driver of intestinal microbiota composition *in vitro* [12].

The dual role of colonic mucus can be explained by the existence of two layers: The inner layer is densely packed, firmly attached to the intestinal epithelium and devoid of bacteria, whereas the outer layer is loose, constantly removed and colonized by bacteria [7, 13]. Colonic mucus is mainly composed of the gel-forming mucin type 2 (MUC2) [14, 15], a heavily O-glycosylated protein secreted by intestinal goblet cells [16–18]. Additionally, mucus contains salts, lipids [19] and defense-related proteins, such as antimicrobial peptides, growth factors, trefoil factors, immunoglobulins and other proteins [13].

Our understanding of colonic mucus structure, function and composition has been largely dependent on *in vivo* models, such as rodents and pigs (reviewed in Etienne-Mesmin *et al.* [20]) or *ex vivo* techniques using human mucosal biopsies (first established by Browning *et al.* [21]) and murine intestinal explants [22, 23]. These models, however, show large heterogeneity between and within subjects or, in the case of animal models, have a poor translational value. Additionally, these models are often expensive, require specialized experience and pose ethical concerns. Attempts have been made to recapitulate the intestinal mucosal layer–including host cells–*in vitro*, varying from simple to more advanced systems. Relatively simple cell models include the use of mucus-excreting colonic cancer cell lines, such as HT29-MTX and LS174T cells [24, 25], but both examples have limitations. HT29-MTX cells have successfully been grown in confluent monolayers [26], but this cell type predominantly secretes MUC5AC, a mucin that is present in the stomach and airways, while producing only a limited amount of colon-specific MUC2 [24]. On the other hand, LS174T cells do produce MUC2 [25], but are not capable of growing in an organized and adherent cell layer [27]. More advanced models include culturing cell lines in innovative models, such as gut-on-chips [28, 29] or 3D scaffolds [30–32]; or the use of human colonoids [33, 34] and human intestinal organoids [35]; or a combination [36], which all demonstrated increased MUC2 production *in vitro* compared to conventional cell culture. Although these models resemble the *in vivo* colonic mucosal layer more closely in terms of mucus composition, they are highly expensive and require specialized expertise [20].

To obtain a more physiologically relevant mucus barrier, simpler yet effective alternative strategies have been shown to further increase mucus production and/or secretion in intestinal cancer cell lines, using biochemical compounds, such as prostaglandin E2 [37] and Notch γ-secretase inhibitors [38], or bacteria-derived compounds (e.g. sodium butyrate [39] and LPS [40]). Physical strategies have also been applied, e.g. growing intestinal porcine epithelial cells at an air-liquid interface (ALI) [41, 42], stimulating cells mechanically, or a combination of both. For instance, Navabi and colleagues managed to create polarized, functional, crypt-forming intestinal cell layers with an adherent mucus layer, when growing HT29-MTX-E12 and other intestinal cell lines on Transwell membranes in semi-wet interfaces with mechanical

stimulation (SWMS). SWMS conditions include decreased apical and basolateral medium volumes and continuous shaking on a rocking platform. Importantly, these cells demonstrated increased expression of MUC2, both absolute and relative to MUC5AC [27]. HT29-MTX-E12 cells grown under these conditions produced a thicker layer compared to static conditions. The positive effect of mechanical stimulation was also shown in more recent *in vitro* models, showing a positive effect on MUC2 production [28]. However, it remains unexplored what molecular mechanisms are involved.

In our study, we aimed to further explore the cellular processes underlying the increased production of MUC2 by HT29-MTX-E12 cells grown under SWMS conditions. To this end, cells were subjected to transcriptome analysis after 15 days of culture to investigate underlying molecular pathways involved. As control cell lines for the transcriptomic analysis, we included Caco-2 and LS174T cells, a non-mucus producing and MUC2-producing cell line, respectively. Next, we further characterized the HT29-MTX-E12 monolayer, by measuring transepithelial electrical resistance and quantifying cell density. Additionally, as similar (semi-wet only) models have shown a more aerobic cell metabolism [43–46], we quantified pH and lactate production per molecule glucose of the conditioned medium.

In brief, our study confirms the upregulation of MUC2 in HT29-MTX-E12 cells cultured under SWMS. Additionally, our study demonstrates upregulation of cell cycle processes, downregulation of KLF4, differential regulation of ion transporters and increased aerobic metabolism of cells cultured under SWMS versus static conditions. In overall, we attempted to gain more insight into the potential mechanisms underlying increased MUC2 production in HT29-MTX-E12 cells grown under SWMS conditions.

## Materials and methods

### Cell culture

HT29-MTX-E12 cells (ECACC) were obtained from Sigma-Aldrich (Darmstadt, Germany). Caco-2 (ATCC HTB-37) and LS174T (ATCC CL-188) cells were purchased at LGC Standards (Wesel, Germany). All cell types were cultured in Dulbecco's Modified Eagle Medium with 4.5g/L glucose, 110 mg/L sodium pyruvate and 584 mg/L L-glutamine (Corning, NY, USA) supplemented with 10% Fetal Bovine Serum and 1% penicillin/streptomycin. When cells reached 80–90% confluency, they were counted and seeded. Passage numbers between 15 and 27 were used for Caco-2 cells and between 3 and 21 for HT29-MTX-E12 cells. LS174T cells were used between passage 17 and 19. Caco-2 and HT29MTX-E21 cells were seeded (day 0) at a density of 273,000 cells/mL in 275 μL per well on 12 mm 0.4 μm-pore polyester Transwell membranes (Corning 3460). A volume of 1 mL medium was added to the basolateral compartment. One day after seeding (day 1), media of all Transwells was refreshed and Semi-Wet conditions with Mechanical Stimulation (SWMS) were applied [27, 32]. To these Transwells, 75 μL and 850 μL of medium was added to the apical and basolateral compartments, respectively, and plates were put on a $CO_2$-resistant shaker (Thermo Fisher Scientific, Breda, The Netherlands) at 65 rpm. Media volumes of Transwells grown under static conditions were unchanged. LS174T cells were seeded at similar seeding density on regular 12-well cell culture plates. This cell line was cultured under static conditions in a regular wells plate only, since these cells do not form a continuous monolayer of cells, but grow in a rather irregular manner [27]. Medium was refreshed every Monday, Wednesday and Friday and cells were harvested 15 days after seeding (t = 15 days). Pictures of the cells on Transwell membranes were taken with a Leica DFC450C microscope camera using Leica Application Suite X software. A 20x magnification was used.

## RNA isolation

Cells were washed with ice-cold PBS twice, trypsinized and RNA was isolated using the Maxwell® 16 LEV simplyRNA Cells Kit (Promega, cat. no. AS1270) and the Maxwell® 16 MDx Instrument (Promega), following the manufacturer's instructions.

## Microarray

RNA isolate from three independent biological replicate experiments was used for microarray analysis. Total RNA yield was measured using photometry (DeNovix, USA). RNA quality was determined on an Agilent 2100 Bioanalyzer (Agilent Technologies, Amsterdam, The Netherlands). RNA was only used when the RNA integrity number (RIN) exceeded 8.0. One hundred nanogram of RNA was converted to cDNA and labelled (Ambion WT expression kit, Life Technologies, Bleiswijk, The Netherlands). Samples were hybridized to an Affymetrix Human Gene 1.1 ST array plate according to the standard Affymetrix instructions (Affymetrix, Santa Clara, CA, USA). The robust multi-array average (RMA) pre-processing algorithm in the Bioconductor library AffyPLM was used to obtain normalized expression estimates [47]. Probe sets were defined and assigned as described by Dai *et al.* [48]. Differences in gene expression between static and SWMS conditions per cell type were analyzed using the Intensity Based Moderated T statistics (IBMT) [49], using $p$ values $<0.05$ as threshold. The Venn diagram was created using Venny 2.1 [50]. Microarray data has been submitted to the Gene Expression Omnibus (GEO) at the NCBI (GSE173729).

## TEER measurements

The transepithelial electrical resistance (TEER) was measured with an EVOM2 Volt/Ohm meter using STX2 electrodes (World Precision Instruments) at day 4, 7, 9, 11 and 14. For each biological replicate (= independent experiment, $n = 3$), two technical replicates (= wells) were measured. One hour prior to measuring, medium of the Transwells was refreshed and equal medium volumes were applied in all wells (275 μL apical and 1 mL basolateral). Before TEER measurements were performed, Transwells were put at room temperature for 5 minutes to allow temperature equilibration. After TEER measurements, the medium volumes in the wells were adapted again to volumes of the respective conditions. The background value (i.e. TEER value of an empty Transwell) was subtracted from the total TEER values.

## FITC-dextran assays

The passage of Fluorescein isothiocyanate–dextran with an average molecular weight of both 4 kDa (FITC-D4) and 40 kDa (FITC-D40) (Sigma-Aldrich, FD4 and FD40) from the apical to the basolateral compartment was measured. FITC-D40 was dissolved in DMEM without phenol red (Gibco, 21063029) at a concentration of 1 mg/mL. A control was added with cells from which the mucus layer was removed. To this end, cells on Transwells were treated for 1 hour with 60 mM N-Acetyl-L-cysteine (Sigma, A9165). After washing with PBS, 275 μL FITC-D40 was added to the apical membrane and 1000 μL DMEM without phenol red to the basolateral membrane. After 3 hours incubation at 37˚C/5% $CO_2$, 100 μL of medium was taken from the basolateral compartment and measured on a Spectramax M2 fluorescence plate reader at 490/ 520 nm (excitation/emission). An empty Transwell served as a control for complete permeability. For each biological replicate (= independent experiment, $n = 3$), two technical replicates (= wells) were measured.

## Protein isolation

Cells were washed twice with PBS and lysed in RIPA Lysis and Extraction Buffer (Thermo-Fisher Scientific), supplemented with the protease and phosphatase inhibitors PhosSTOP and cOmplete (Roche Diagnostics, Almere, The Netherlands). Lysates were incubated on ice for 20 minutes following centrifugation for 10 minutes at 13,000 $x\,g$. Supernatant was collected and protein concentrations were measured using a bicinchoninic acid assay (Thermo Fisher Scientific).

## Western blot

Protein lysates (20 μg of protein/lane) were loaded onto 4–15% Mini-PROTEAN TGX Precast Protein Gels (Bio-Rad, Veenendaal, The Netherlands). Next, proteins were transferred onto a polyvinylidene difluoride membrane (Trans-Blot Turbo Midi 0.2 μm PVDF Transfer Packs, Bio-Rad) using the Transblot Turbo System (Bio-Rad). Membranes were blocked for 1 hour and incubated overnight at 4˚C with rabbit anti-KLF4 (Sigma-Aldrich, catalogue no. SAB1300678) and rabbit anti-HSP90 (Cell Signaling Technology, cat. no. 4874). Antibodies were used in 1:500 and 1:5000 dilutions for KLF4 and HSP90, respectively. Membranes were incubated for 1 hour with goat anti-rabbit (GenScript, cat. no. A00098) diluted 1:5000. Blocking and incubation of primary and secondary antibodies were done in TBS with 0.1% Tween 20 (TBS-T) and 5% (w/v) skimmed milk. In between, membranes were washed in TBS-T. Signals were quantified using the ChemiDoc MP system (Bio-Rad) and Clarity ECL substrate (Bio-Rad).

## Dot blot

Proteins were diluted to 68.2 μg/mL and 6x serially diluted 1:2 in PBS. In total 7 dilutions and one PBS-control were blotted per condition. Of each dilution, 50 μL of sample was blotted on a Pierce 0.2 μm nitrocellulose membrane (Thermo Scientific) in a Dot Blot device (The Convertible, cat. series 1055. Gibco BRL) connected to a vacuum-pump. Next, membranes were blocked, incubated and imaged as described for Western Blot. Primary antibodies were diluted 1:2000 and 1:1000 for mouse anti-MUC2 (Abcam, cat. no. ab11197) and-MUC5AC (Sigma-Aldrich, cat. no. WH0004586M7) respectively. Secondary goat anti-mouse antibody (Genscript, cat. no. A00160) was diluted at 1:2500. To check protein quantity on the membranes, a separate, identical membrane was blotted and incubated for 10 minutes in Ponceau Red (Honeywell Fluka). Next, the blot was washed in demi-water and signals were quantified using the ChemiDoc MP system. Dot Blots with anti-MUC2 and–MUC5AC were quantified by measuring the density of the first six dots (rows) using ImageJ software. A density-based linear trendline was calculated from the second to the sixth dot. The coefficients (per μg/mL) were corrected for Ponceau Red density. Ponceau Red signals were quantified by measuring density of the first dot (first row) only. Biological replicates per condition were $n = 3$.

## pH, lactate and glucose measurements

Every time medium of Transwells was refreshed, conditioned medium was collected from HT29-MTX-E12 cells and stored at -20˚C until further processing. Medium of apical and basolateral compartments were collected separately, but compartments were pooled per time point and condition. Samples were spun down and 200 μL of supernatant was transferred to a 96-wells plate. After stabilization, absorbance was measured at 415 and 560 nm at 5% $CO_2$ in a Synergy Neo2 Hybrid Multi-Mode Reader with a $CO_2$ and $O_2$ gas controller (1210013) (Bio-Tek Instruments). Aliquots of growth medium (ca. 5 mL) were used for preparation of a

standard curve with pH values ranging from 2.6 to 9.9. Lactate and glucose were quantified with a Shimadzu LC2030C-Plus high-performance liquid chromatography (HPLC) system equipped with a Shodex SH1821 column kept at 45˚C and running 0.01 N sulfuric acid as eluent (1 mL/min). Compounds were detected by determining the refractive index and identified using pure lactate and glucose as external standards and crotonate as an internal standard.

### Statistical analysis

Statistical analysis of microarray data is described above. Data distribution for other (continuous) outcomes was tested with the D'Agostino-Pearson omnibus normality test using Graph-Pad Prism (San Diego, CA, USA). Significance of differences between two conditions was analysed using a two-tailed, unpaired Student's t-test. For non-normally distributed data, a Mann-Whitney U test was used. The data are presented as mean ± standard deviation. A $p$-value of $\leq 0.05$ was considered significant. All experiments were performed in three biological replicates.

## Results

### Gene expression changes in HT29-MTX-E12, Caco-2 and LS174T cells

To investigate SWMS-specific effects leading to higher mucus production of the SWMS culturing method, HT29-MTX-E12 cells were grown for 14 days under static and SWMS conditions (after cell seeding and attachment), and microarray analysis was performed on mRNA of the cells. We included Caco-2 cells, a non-mucus producing cell line, grown on Transwell membranes under static and SWMS conditions for 14 days as a reference control. The LS174T cell line was used as a control for a mucus-producing cell line. Multilevel principal component analysis performed on the 500 most variable genes showed a strong clustering of the biological replicates per cell type with a clear separation of the three cell lines (Fig 1A). A number of 6,732 and 7,013 genes were differentially regulated between SWMS and static conditions in HT29-MTX-E12 and Caco-2 cells, respectively ($p < 0.05$) (Fig 1B and S1A and S1B Fig).

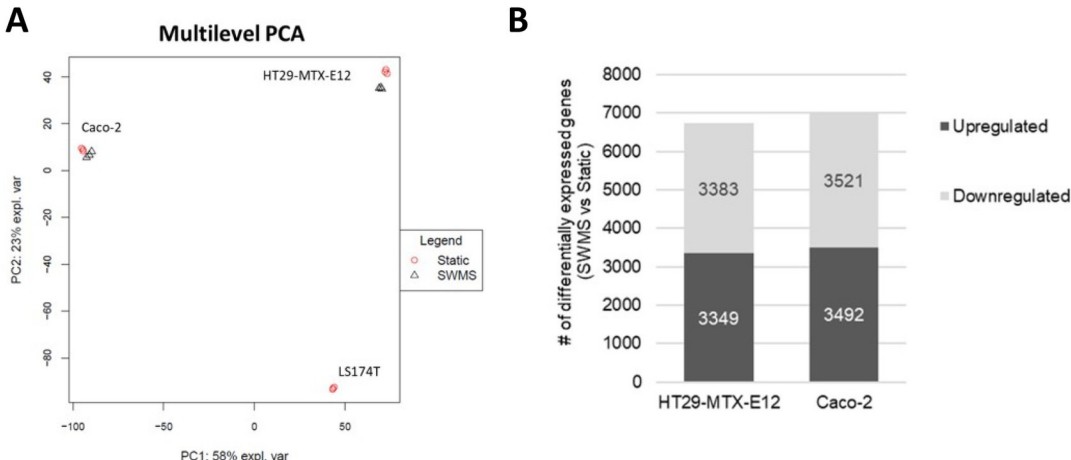

**Fig 1. Overview microarray results in HT29-MTX-E12, Caco-2 and LS174T cells grown under static and/or SWMS conditions. A)** Multilevel principal component analysis (PCA) of the 500 most variable genes. **B)** Number of differentially expressed genes between SWMS and static conditions in HT29-MTX-E12 and Caco-2 cells (IBMT $p < 0.05$). Microarray results are based on $n = 3$ biological replicates per condition.

## Increased MUC2/MUC5AC ratio in SWMS-cultured HT29-MTX-E12 cells

We investigated the expression of genes encoding for mucins in HT29-MTX-E12 cells, since we know from previous studies that culturing this cell type under SWMS conditions led to increased MUC2 production [27]. Indeed, in our study, SWMS conditions resulted in a significant 1.42-fold upregulation of *MUC2* ($p < 0.05$) (Fig 2A). Expression of gel-forming mucins *MUC5AC* and *MUC5B* was decreased in response to SWMS (FC = -1.56 and -1.84, respectively, $p < 1 \cdot 10^{-4}$) (Fig 2A, S1 File). Transmembrane mucins were also significantly lower expressed after SWMS conditions (Fig 2A). *MUC3A*, *13*, *17*, *20* and *21* showed FC values between -1.52 and -2.32 (S1 File). Eight out of 19 mucins (*MUC4*, *6*, *7*, *12*, *15*, *16*, *19* and *22*) displayed very low expression levels under both static and SWMS conditions (RMA < 4) and were not significantly differentially expressed (S1 File). Taken together, these data indicate that SWMS conditions resulted in a change in expression of mucin encoding genes with an increased expression of *MUC2*, while other gel-forming as well as transmembrane mucins were significantly lower or not differentially expressed.

Dot Blot data supported microarray data for MUC2, as protein levels also showed a significant upregulation ($p < 0.05$) under SWMS versus static conditions (Fig 2B, S2A and S2C Fig). Less MUC5AC was detected under SWMS conditions, however, variation between replicates was rather high and differences were not significant ($p = 0.20$) (Fig 2B and S2B and S2C Fig). Overall, HT29-MTX-E12 cells grown under SWMS conditions showed a significantly increased MUC2/MUC5AC-ratio (Fig 2C, $p < 0.05$), confirming previous findings [27]. For LS174T, which we included as a control cell line reported to produce predominantly MUC2, a relatively high MUC2/MUC5AC ratio was calculated (S2D and S2E Fig).

In contrast, Caco-2 cells, which are not reported to produce mucus, showed low RMA values (< 5) for *MUC2* and *MUC5AC* and protein expression of these mucins was below detection level in both conditions (S2F and S2G Fig). Furthermore, similar to HT29-MTX-E12 cells, SWMS conditions resulted in significant downregulation of *MUC20*, *MUC3A* and *MUC13* in this cell line (S1 File).

## Upregulation of genes and pathways related to cell cycle, cell growth and cell proliferation in cells cultured under SWMS conditions

As both HT29-MTX-E12 and Caco-2 cell lines were subjected to SWMS conditions and grown under static conditions, we compared the genes up- or downregulated in both cell lines. Of the 2,844 significantly differentially expressed genes in both cell lines, 353 genes (12.4% of shared total significantly regulated genes) were upregulated by 1.5-fold or higher in both cell types due to the SWMS conditions. A total of 180 genes (6.3%) were significantly downregulated by 1.5-fold or higher in both cell lines and a total of 98 genes showed opposite effects when comparing the two cell lines (S3 Fig). Interestingly, among the shared upregulated genes, a number of cell cycle related genes was included. Moreover, the top 10 most upregulated genes of HT29-MTX-E12 cells under SWMS versus static conditions was dominated by genes related to cell growth, cell motility and cell proliferation, i.e. *KIF14*, *KIF20A*, *DEPDC1*, *DLGAP5* and *SPC25* (Table 1).

Gene Set Enrichment Analysis (GSEA) also revealed that most significantly enriched upregulated pathways were dominated by cell cycle and DNA replication related pathways (Table 2 and S2 File). Furthermore, marker of proliferation Ki-67 (*MKI67*) was 5.03-fold upregulated by SWMS culturing conditions ($p < 1 \cdot 10^{-10}$) in HT29-MTX-E12 and 2.33-fold in Caco-2 cells ($p < 1 \cdot 10^{-5}$) (S4A Fig). Additionally, we found higher cell counts at day 15 under SWMS compared to static conditions in both cell lines (S4B Fig), confirming that SWMS conditions lead to an increase in cell proliferation. Closely related to cell cycle genes are

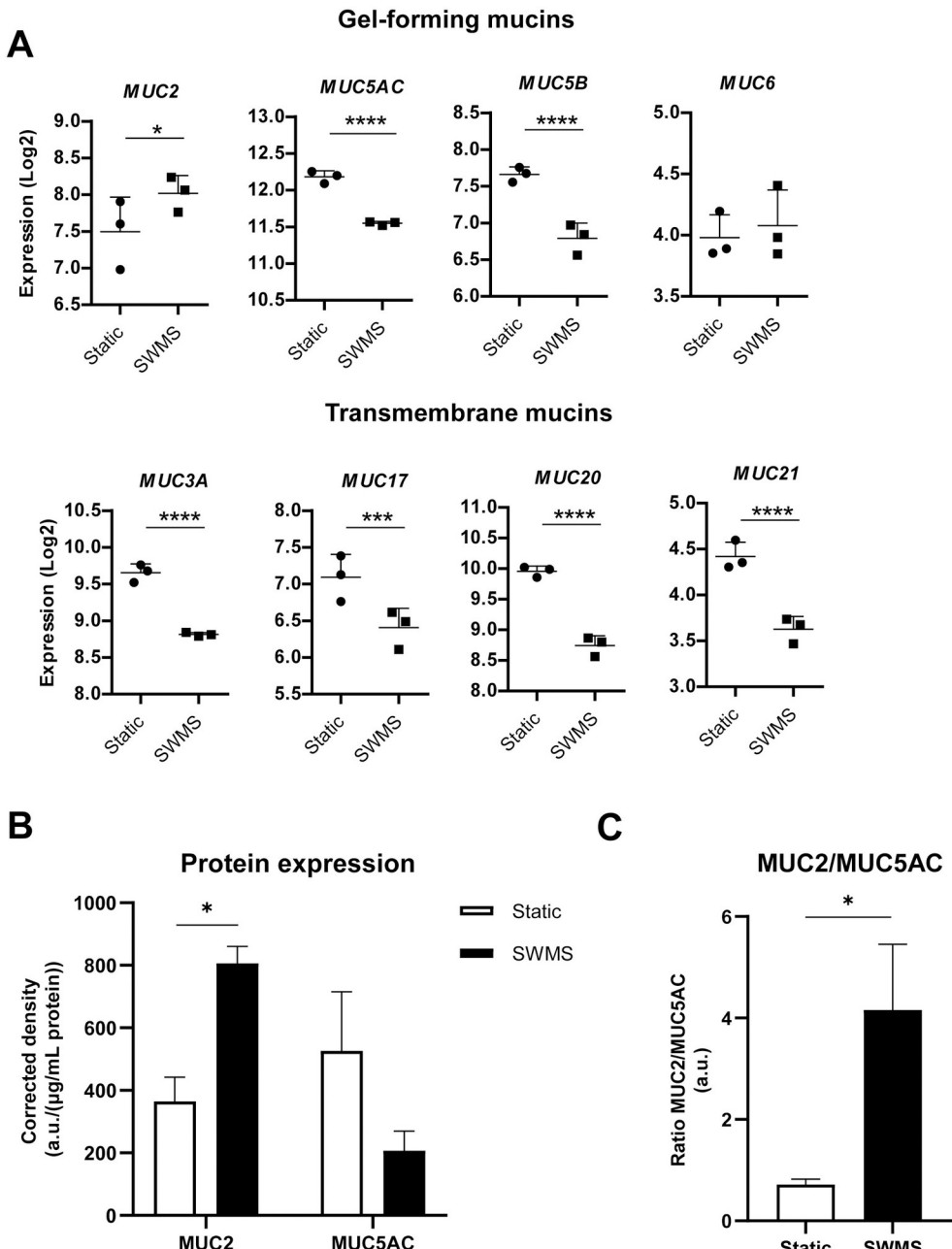

**Fig 2. Mucin gene and protein expression in HT29-MTX-E12 cells grown under static and SWMS conditions. A)** Microarray gene expression values (Log2) of a panel of secretory and transmembrane mucins **B)** Protein expression of MUC2 and MUC5AC, expressed as density (a.u.) per µg/mL protein blotted, after correction of Ponceau Red density. Blot data in S2 Fig) **C)** Ratio of MUC2 and MUC5AC protein expression. * $p < 0.05$; *** $p < 0.001$; **** $p < 0.0001$; $n$ = 3 biological replicates.

cytoskeleton-related genes, some of which also showed differential regulation under SWMS (S1 File). Altogether, these data indicate that SWMS conditions result in the activation of cell proliferation pathways compared to static conditions, independent of cell type.

**Table 1. Top 10 strongest up- and downregulated genes in HT29-MTX-E12 cells grown under SWMS versus static conditions.**

**UPREGULATED**

| Gene | Fold Change | p-value | Gene name |
|------|-------------|---------|-----------|
| RNY4P23 | 10.80 | 5.40E-06 | RNY4 pseudogene 23 |
| SUCNR1 | 8.78 | 2.55E-11 | succinate receptor 1 |
| KIF14 | 8.00 | 2.45E-12 | kinesin family member 14 |
| KIF20A | 7.99 | 1.65E-08 | kinesin family member 20A |
| DEPDC1 | 7.93 | 6.11E-11 | DEP domain containing 1 |
| DLGAP5 | 7.91 | 5.15E-12 | DLG associated protein 5 |
| HMMR | 7.82 | 1.03E-15 | hyaluronan mediated motility receptor |
| H4C1 | 7.56 | 7.27E-07 | H4 clustered histone 1 |
| IL33 | 7.44 | 2.98E-13 | interleukin 33 |
| SPC25 | 7.41 | 4.47E-13 | SPC25 component of NDC80 kinetochore complex |

**DOWNREGULATED**

| Gene | Fold Change | p-value | Gene name |
|------|-------------|---------|-----------|
| TFF2 | -15.75 | 2.62E-13 | trefoil factor 2 |
| SCGB2A1 | -8.91 | 1.33E-10 | secretoglobin family 2A member 1 |
| ST8SIA6 | -4.56 | 1.06E-08 | ST8 alpha-N-acetyl-neuraminide alpha-2,8-sialyltransferase 6 |
| DUOX2 | -4.53 | 4.52E-09 | dual oxidase 2 |
| TFF1 | -4.44 | 1.37E-16 | trefoil factor 1 |
| APOL1 | -4.41 | 1.29E-06 | apolipoprotein L1 |
| ADGRF1 | -3.96 | 7.06E-08 | adhesion G protein-coupled receptor F1 |
| DUOXA2 | -3.88 | 1.67E-09 | dual oxidase maturation factor 2 |
| UPK3B | -3.80 | 4.29E-09 | uroplakin 3B |
| SPRR1B | -3.63 | 2.31E-07 | small proline rich protein 1B |

## Expression of target genes involved in Notch- and Atoh-key pathways did not point towards a favoured cell differentiation state by SWMS conditions

Next to the marked effects of SWMS conditions on cell proliferation in both HT29-MTX-E12 and Caco-2 cells, we took a closer look at the effects on intestinal cell differentiation. Key regulators in this process belong to the Notch/Atoh signalling pathway [51, 52]. When it comes to cell differentiation, Notch and Atoh have opposing roles: Notch activation promotes

**Table 2. Top 10 most significantly enriched upregulated pathways in HT29-MTX-E12 cells induced by SWMS conditions.**

| Pathway entry | Enriched upregulated pathways | NES* | FDR q-value |
|---------------|-------------------------------|------|-------------|
| HSA04110 | Cell Cycle | 2.856 | 0.000 |
| HSA03460 | Fanconi Anemia Pathway | 2.637 | 0.000 |
| HSA03030 | DNA Replication | 2.596 | 0.000 |
| HSA05322 | Systemic Lupus Erythematosus | 2.539 | 0.000 |
| HSA03440 | Homologous Recombination | 2.506 | 0.000 |
| HSA04114 | Oocyte Meiosis | 2.329 | 0.000 |
| HSA03420 | Nucleotide Excision Repair | 2.315 | 0.000 |
| HSA03430 | Mismatch Repair | 2.312 | 0.000 |
| HSA00100 | Steroid Biosynthesis | 2.306 | 0.000 |
| HSA04914 | Progesterone Mediated Oocyte Maturation | 2.268 | 0.000 |

*NES: Normalised enrichment score.

absorptive cell differentiation, while Atoh activation favours differentiation into secretory cell types [51–53]. Interestingly, in our transcriptomic dataset, both *NOTCH1* and *ATOH1* were significantly upregulated by SWMS conditions in HT29-MTX-E12 cells (FC = 1.61, $p < 0.001$ and FC = 1.64, $p < 0.01$, respectively) and the latter also in Caco-2 cells (FC = 2.09, $p < 0.001$). The stem cell marker *LGR5*, which is known to be part of a positive feedback loop regulated by Notch [54], was significantly upregulated by SWMS conditions in HT29-MTX-E12 cells (FC = 2.33, $p < 1 \cdot 10^{-5}$), although RMA values were below 5. In Caco-2 cells, *LGR5* had higher RMA values and was even stronger upregulated (FC = 4.37, $p < 1 \cdot 10^{-7}$). Further downstream the Notch pathway, the target genes Hairy and enhancer of split (HES) family members *HES1* and *HES6* were both significantly upregulated by SWMS conditions in HT29-MTX-E12 cells (FC = 1.24, $p < 0.05$ and FC = 2.08, $p < 1 \cdot 10^{-5}$, respectively). Apart from Cyclin D1 (*CCND1*) (FC = -1.51, $p < 1 \cdot 10^{-6}$) no other Notch-target genes were significantly differentially expressed. Regarding the Atoh pathway, the downstream target gene Neurogenin 3 (*NEUROG3*) was 2.07-fold and 2.41-fold higher expressed ($p < 0.001$) in HT29-MTX-E12 and Caco-2 cells, respectively. Other Atoh target genes, such as *SPDEF* and *GFI1* were not differentially expressed between SWMS and static conditions in both cell types. Altogether, these transcriptomic data show that some target genes of both the Notch and Atoh pathway were differentially expressed in HT29-MTX-E12 and Caco-2 cells. However, these data do not point towards a favoured differentiation state (absorptive versus secretory cell fate).

## Downregulation of KLF4 at both gene and protein level under SWMS conditions in HT29-MTX-E12 and Caco-2 cells

Another Notch-target involved in differentiation of progenitor cells into goblet cells is KLF4 [55]. Given the secretory, goblet cell-like phenotype of HT29-MTX-E12 cells grown under SWMS conditions, the significant downregulation of *KLF4* (FC = -1.86, $p < 0.0001$) is interesting. Acting as both a transcriptional repressor and activator in the gastrointestinal epithelium, this zinc-finger transcription factor plays a critical role in the decision between proliferation and cell cycle arrest/differentiation [56]. We validated the downregulation of KLF4 under SWMS conditions at protein level using Western Immunoblotting. Indeed, when cultured under SWMS conditions, the expression of KLF4 protein was lower compared to the static conditions in both cell lines (Fig 3A and 3B and S5 Fig). We next investigated whether this downregulation of KLF4 resulted in differential expression of target genes of KLF4, as identified in literature [57]. Among the most strongly regulated are genes related to cell-cycle control or essential for cell proliferation and differentiation, including cyclin B1 (FC = 6.59, $p < 1 \cdot 10^{-13}$) [58], cyclin E2 (FC = 2.03, $p < 0.001$) [59, 60], ornithine decarboxylase (FC = 1.80, $p < 1 \cdot 10^{-7}$) [61], cyclin E1 (FC = 1.62, $p < 1 \cdot 10^{-4}$) [59, 60], cyclin D1 (FC = -1.51, $p < 1 \cdot 10^{-6}$) [62, 63] and cyclin D2 (FC = 1.85, $p < 0.001$) [64]. A non-exhaustive list of genes (in) directly related to KLF4 is provided in S1 File. In contrast to *KLF4*, other reported goblet-cell markers such as *AGR*, *CDX2*, *MEP1β* and *FCGBP*, amongst others, were not significantly regulated in HT29-MTX-E12 cells grown under SWMS. A non-exhaustive list of "Goblet cell markers" [65] is given in S1 File.

## Minor effect on transepithelial electrical resistance, but increased paracellular permeability of HT29-MTX-E12 cells grown under SWMS conditions

To investigate the effects of SWMS conditions on intestinal epithelial barrier integrity, transepithelial electrical resistance (TEER) was measured at multiple time points during culturing. TEER values of HT29-MTX-E12 cells increased steadily over time and no clear effects were

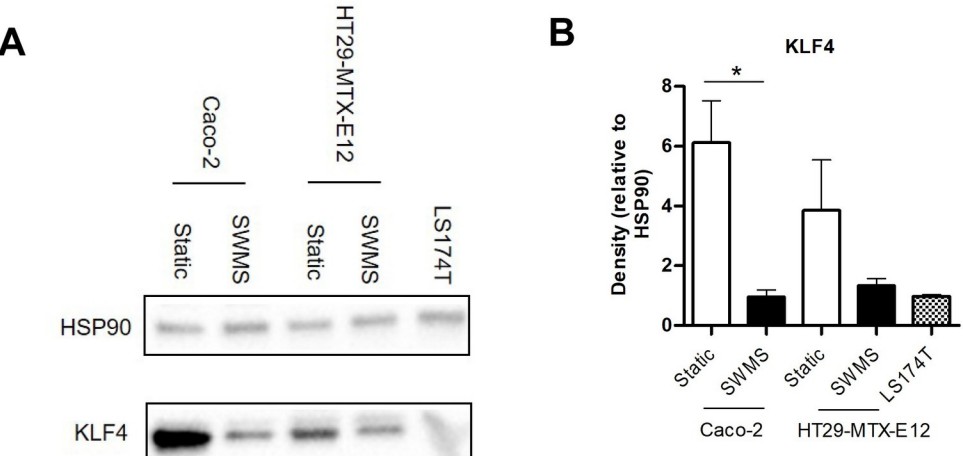

**Fig 3. Protein expression of KLF4 in HT29-MTX-E12, Caco-2 and LS174T cells grown under static and/or SWMS conditions. A)** Western Blot results of KLF4 and HSP90 (house-keeping protein) in Caco-2, HT29-MTX-E12 and LS174T-cells grown under static and SWMS conditions or static only (LS174T). Of each condition, one representative biological replicate is shown. See S5 Fig for all replicates. **B)** KLF4 protein quantity expressed as the protein band density relative to HSP90. * $p < 0.05$, $n$ = 3 biological replicates.

observed between static and SWMS conditions (Fig 4A). For Caco-2 cells, the TEER of the cells grown on SWMS conditions increased steeply during the first seven days of culturing and decreased gradually afterwards, while the TEER of cells grown on static conditions was relatively stable (Fig 4B). For HT29-MTX-E12, we complemented TEER data with assays using fluorescent probes (FITC-dextran of 4 and 40 kDa) to quantify paracellular permeability. Interestingly, at t = 14 days, HT29-MTX-E12 cells grown under SWMS showed higher permeability than static conditions for both probe sizes (Fig 4C and 4D). After washing off the mucus with a mucolytic reagent (N-Acetyl-L-cysteine (NAc)), however, no differences in paracellular permeability could be observed. Based on available literature, we compiled a panel of genes responsible for epithelial barrier integrity (S1 File). Except for a 3.33-fold increase in expression of Catenin alpha like 1 (*CTNNAL1*) in HT29-MTX-E12 cells under SWMS conditions ($p < 1 \cdot 10^{-9}$), only a low number of significantly differentially expressed genes was found for both HT29-MTX-E12 and Caco-2 cells (S1 File). Together, these results indicate that SWMS conditions had negligible effects on TEER in HT29-MTX-E12 cells, which was reflected by the relatively low number of significantly differentially expressed genes related to cell integrity, but increased paracellular permeability. Interestingly, a clear difference in cell appearance on the Transwell membranes was observed for both cell types at the time cells were harvested, as cells grown under SWMS conditions seemed to concentrate in the center of the membrane (S6A and S6B Fig).

## Downregulation of trefoil factor genes in HT29-MTX-E12 cells grown under SWMS conditions

When focusing on the most significantly downregulated genes in HT29-MTX-E12 grown under SWMS compared to static conditions, Trefoil Factor 2 (*TFF2*) was most strongly downregulated (FC = -15.75, $p < 1 \cdot 10^{-12}$) (Table 1). Although to a lesser extent, *TFF1* and *TFF3* were also significantly downregulated (Fig 5). Trefoil factors are small cysteine rich peptides and form a family of mucin-associated secretory molecules involved in many physiological processes to maintain and restore gastrointestinal mucosal homeostasis (Reviewed in Aihara

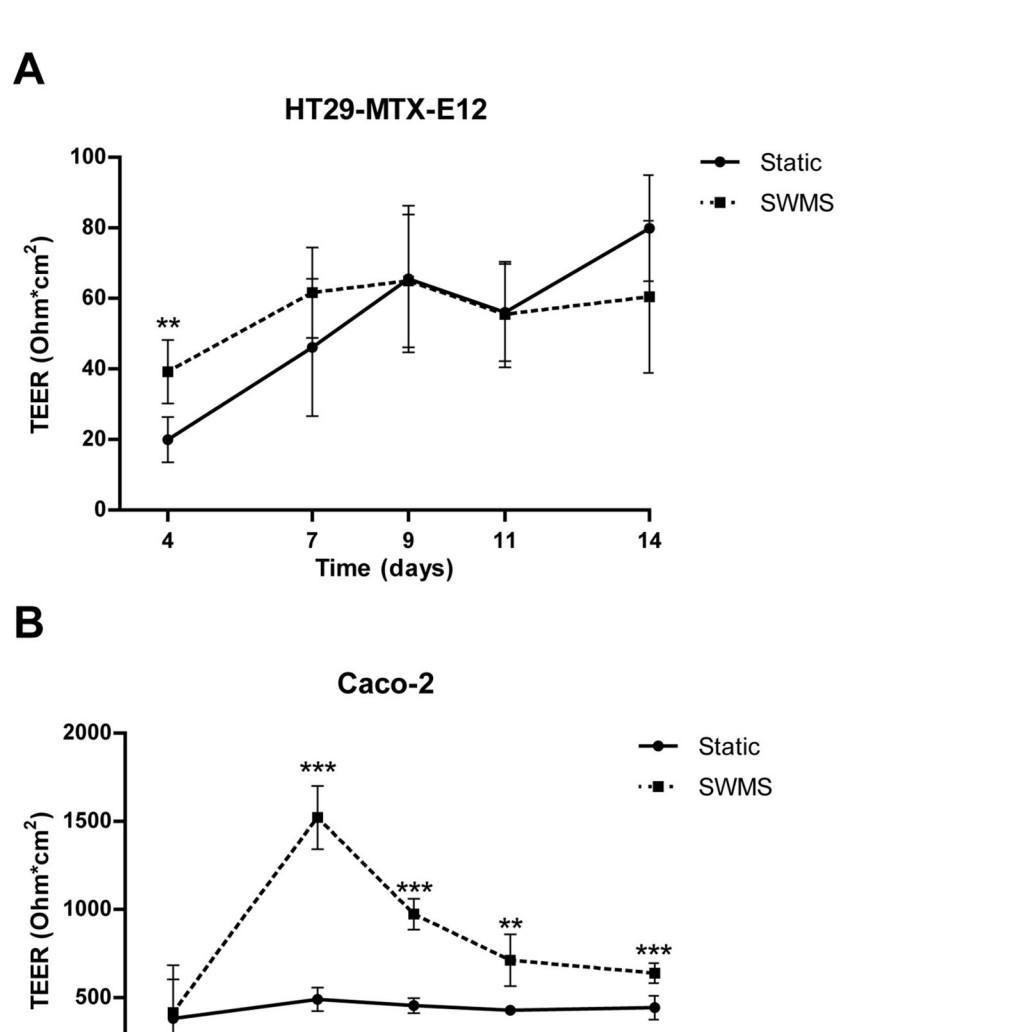

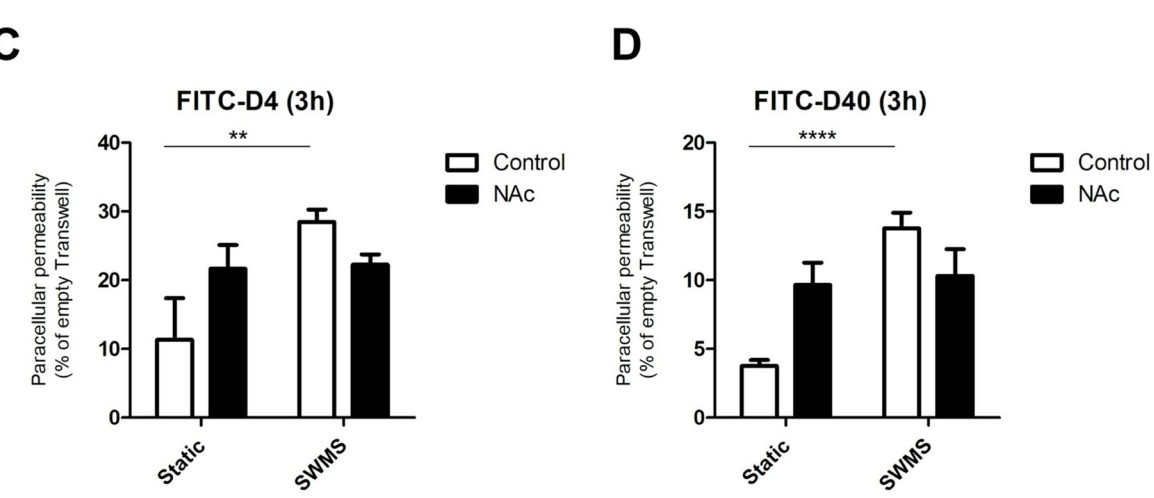

**Fig 4. Transepithelial electrical resistance (TEER) and paracellular permeability of HT29-MTX-E12 and/or Caco-2 cells grown under static and SWMS conditions up to 14 days.** TEER was measured at t = 4, 7, 9, 11 and 14 days and expressed in Ohm/cm$^2$ in **A)** HT29-MTX-E12 and **B)** Caco-2 cells, $n$ = 3 biological replicates. Paracellular permeability was quantified by incubating for 3 hours with FITC-dextran of size **C)** 4 kDA and **D)** 40 kDa. Values were normalized to an empty Transwell, $n$ = 3 biological replicates. Control = with mucus, NAc = treated with N-acetyl-L-cysteine to remove mucus; * $p < 0.05$; ** $p < 0.01$; *** $p < 0.001$.

*et al.* [66]). TFF peptides are known to auto- and cross-regulate their expression via the epidermal growth factor receptor *in vitro* [66, 67]. However, *EGF1* was not significantly differentially expressed between culture conditions in HT29-MTX-E12 cells in our dataset. On the contrary, the gene encoding Epidermal growth factor receptor (*EGFR*) was significantly downregulated by 1.42-fold ($p < 0.0001$).

## Different regulation of ion transporters under SWMS conditions in both HT29-MTX-E12 and Caco-2 cells

Since ion transport has proven crucial in (intestinal) mucus production [68], we explored the effect of SWMS conditions on the expression of ion transporters and exchangers. The gene encoding Cystic Fibrosis transmembrane regulator (CFTR), a chloride and bicarbonate transporter expressed on the apical side of (intestinal) epithelial cells and demonstrated to be indispensable for normal mucus production [69], was significantly upregulated in HT29-MTX-E12 cells grown under SWMS conditions (FC = 1.68, $p = < 0.00001$). Other ion transporters that were significantly regulated under SWMS conditions in HT29-MTX-E12 and Caco-2 cells include *NBCe1/SLC4A4*, encoding a basolateral Na$^+$/HCO3$^-$ importer (FC = -1.95, $p < 0.01$ in HT29-MTX-E12) and *NHE1/SLC9A1*, encoding a basolateral Na$^+$/H$^+$ exchanger (FC = -1.59, $p < 1 \cdot 10^{-5}$). Additionally, the significant downregulation of *DRA/SLC26A3*, encoding an apical Cl$^-$/HCO$_3^-$ exchanger (FC = -1.25, $p < 0.05$) under SWMS conditions is interesting, given its importance in intestinal salt and fluid absorption [70]. This gene was, however, not highly expressed in HT29-MTX-E12 cells (RMA < 3). Additionally, the major regulator of intestinal pH, NHE3 (SLC9A3) [71–73], was however, relatively low expressed in static HT29-MTX-E12 cells (RMA < 6) and only slightly altered under SWMS conditions. Whereas GSEA revealed that the pathway "Mineral absorption" was not significantly regulated in HT29-MTX-E12,

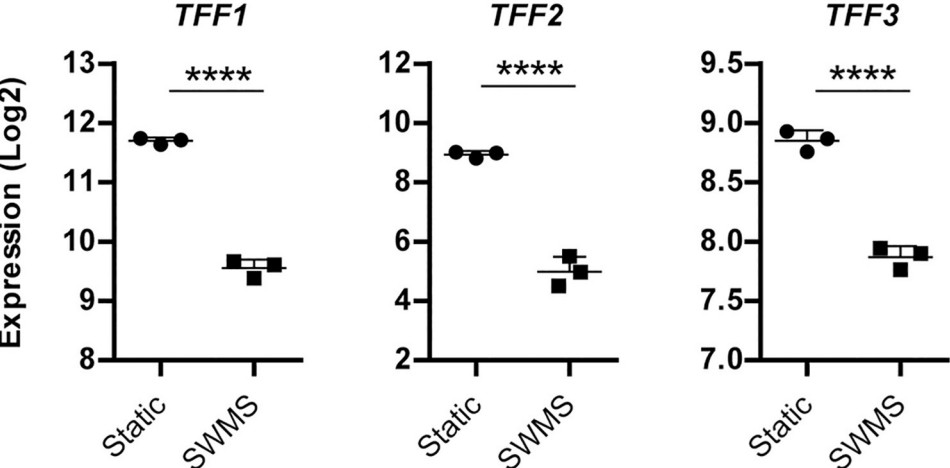

**Fig 5. TFF gene expression in HT29-MTX-E12 cells grown under static and SWMS conditions.** Microarray expression values (Log2) of *TFF1*, *2* and *3* and the corresponding fold change (FC). **** $p < 0.0001$; $n$ = 3 biological replicates.

interestingly, this pathway was the most enriched among all downregulated pathways in Caco-2 cells (NES = -2.58, FDR q-value = 0.00, S2 File). These cells also showed relatively strong and significant regulation of aforementioned ion transporters (FC = 1.78, $p < 1 \cdot 10^{-8}$ for *CFTR*, FC = -2.03, $p < 1 \cdot 10^{-6}$ for *NBCe1*, FC = -1.33, $p < 0.01$ for *NHE1* and FC = -3.18, $p < 1 \cdot 10^{-8}$ for *DRA*). All in all, culturing under SWMS resulted in significant regulation of several key ion transporters in both HT29-MTX-E12 and Caco-2 cells.

### Lower glucose consumption and lactate production per cell in HT29-MTX-E12 cells grown under SWMS conditions

The observed visible colour difference of cell medium in HT29-MTX-E12 cells, accompanied by significant regulation of $H^+$ and $HCO^-$ transporters, could indicate a difference in cell medium pH between growth conditions. The pH of the medium samples collected during every refreshment remained similar between the apical and basolateral compartment in both conditions. During the first 13 days, pH decreased to a similar extent in both conditions. Under SWMS conditions, the pH remained relatively stable over time after one week, whereas static conditions showed a decreased pH at t = 14 days in the apical compartment (difference of 0.2) (S7A and S7B Fig). At t = 15 days, one day after the last medium refreshment, trends reversed and medium of static conditions showed a significantly higher pH compared to SWMS conditions in both compartments (difference of 0.2) (Fig 6A and 6B and S7A, S7C and S7D Fig). A decrease in pH could be explained by increased lactate production and subsequent acidification of the medium. Indeed, pH of cell medium showed opposite trends to the amount of lactate measured in the medium, i.e. lactate concentrations increased with decreasing pH values and remained stable for SWMS conditions (S8A Fig). As medium volumes were, however, different between static and SWMS conditions, we calculated total lactate production and glucose consumption per well. Cell medium collected from cells grown under static

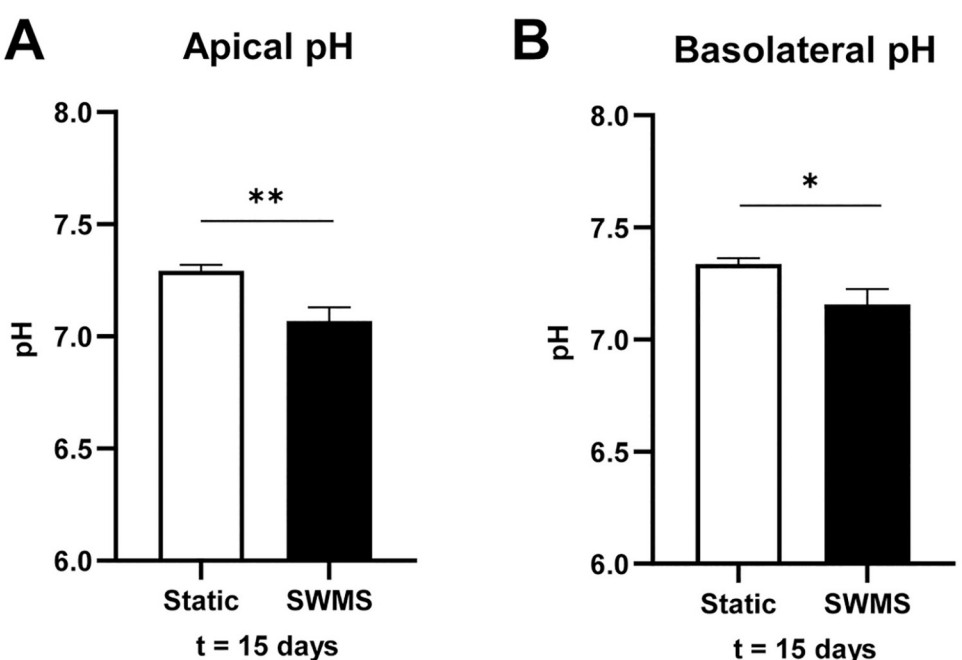

**Fig 6. pH values of medium collected at t = 15 days from HT29-MTX-E12 cells grown under static and SWMS conditions.** Medium pH of **A)** apical and **B)** basolateral compartments of HT29-MTX-E12 cells grown under static and SWMS conditions at t = 15 days. * $p < 0.05$; ** $p < 0.01$, $n = 3$ biological replicates.

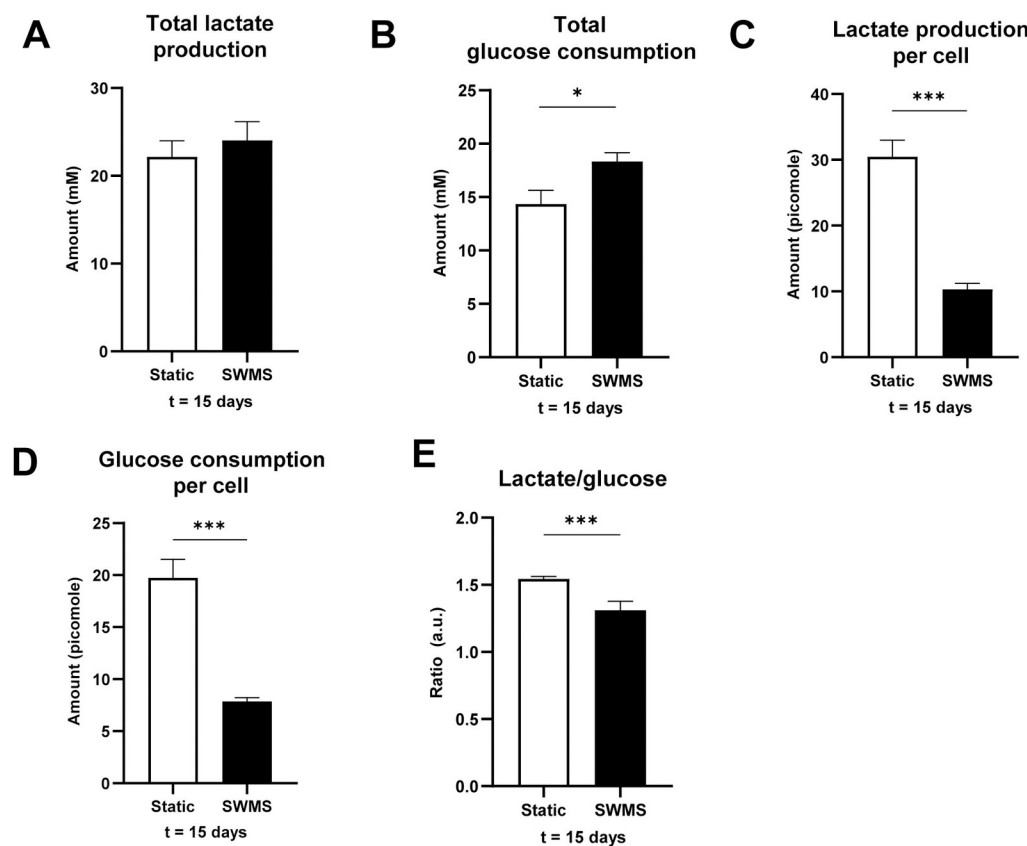

**Fig 7. Lactate production and glucose consumption by HT29-MTX-E12 grown under static and SWMS conditions.**
**A)** Total lactate (micromole) produced per well in medium collected from apical and basolateral compartments of HT29-MTX-E12 cells grown under static or SWMS conditions, at t = 15 days. **B)** Total glucose (micromole) consumed per well from medium collected from apical and basolateral compartments of HT29-MTX-E12 grown under static and SWMS conditions, at t = 15 days. **C)** Glucose consumption (picomole) per cell by HT29-MTX-E12 grown under static and SWMS conditions at t = 15 days. **D)** Lactate production (picomole) per cell by HT29-MTX-E12 grown under static and SWMS conditions at t = 15 days. **E)** Ratio of lactate produced glucose consumed per cell in HT29-MTX-E12 grown under static and SWMS conditions at t = 15 days. * $p < 0.05$; ** $p < 0.01$; *** $p < 0.001$, n = 3 biological replicates.

conditions showed significantly higher lactate production at t = 7–14 days (S8B Fig), accompanied with no difference in total glucose consumption per well (S8C Fig), indicating a lower amount of lactate produced per mole glucose under SWMS conditions. At t = 15 days, one day after the last medium change (at which not all glucose had been consumed yet), lactate production was similar between conditions (Fig 7A). On the contrary, cells grown under SWMS conditions had consumed significantly more glucose (Fig 7B), again resulting in a lower lactate-per-glucose ratio under SWMS conditions. After correction for cell count at t = 15 days, glucose consumption and lactate production per cell were still significantly lower under SWMS conditions (Fig 7C and 7D, $p < 0.001$), resulting in a significantly lower lactate-per-glucose ratio (Fig 7E). Lower glucose consumption under SWMS conditions coincided with a significantly lower expression of *GLUT1/SLC2A1*, encoding a transmembrane glucose transporter (FC = -1.46, $p < 1 \cdot 10^{-7}$) and *HK2*, encoding the glycolytic enzyme hexokinase 2 (FC = -2.18, $p = p < 1 \cdot 10^{-8}$). GSEA revealed, however, no significant enrichment of the "Carbohydrate absorption pathway" or "Glycolysis pathway" in HT29-MTX-E12 cells cultured under SWMS conditions. Interestingly, however, both pathways were among the top-10 enriched downregulated in Caco-2 cells (NES "Carbohydrate absorption" = -2.35, $p = 0.00$ and NES "Glycolysis

pathway" = -2.12, $p < 1 \cdot 10^{-3}$, respectively). As shown previously for similar models, these data could point towards a more aerobic cell metabolism under SWMS conditions [42–46]. This is further supported by a significant downregulation of the "HIF1-signalling pathway" in both HT29-MTX-E12 and Caco-2 cells, as revealed by GSEA (NES = -1.79, FDR q-value = < 0.05 and NES = -2.30, FDR q-value = 0.01, respectively).

## Discussion

In the present study, we aimed to further characterize the potential mechanisms underlying the increased MUC2 production by the SWMS culture method as described by Navabi *et al.* [27]. To this end, we cultured HT29-MTX-E12 cells under both static and SWMS conditions and performed microarray analysis to investigate changes in gene expression by taking both a targeted and untargeted approach. First, we aimed to validate the increased MUC2 production under SWMS conditions. Indeed, SWMS conditions induced higher *MUC2* expression in HT29-MTX-E12 cells, which was also reflected at the protein level. It seemed that the increase in MUC2 occurred at the expense of MUC5AC, since both gene and protein expression of this mucin was decreased. This resulted in a significantly increased MUC2/MUC5AC ratio, confirming previous findings [27]. An overview of the most prominent changes observed at t = 15 days is graphically summarized (Fig 8).

**Effects SWMS vs. static at t = 15 days**

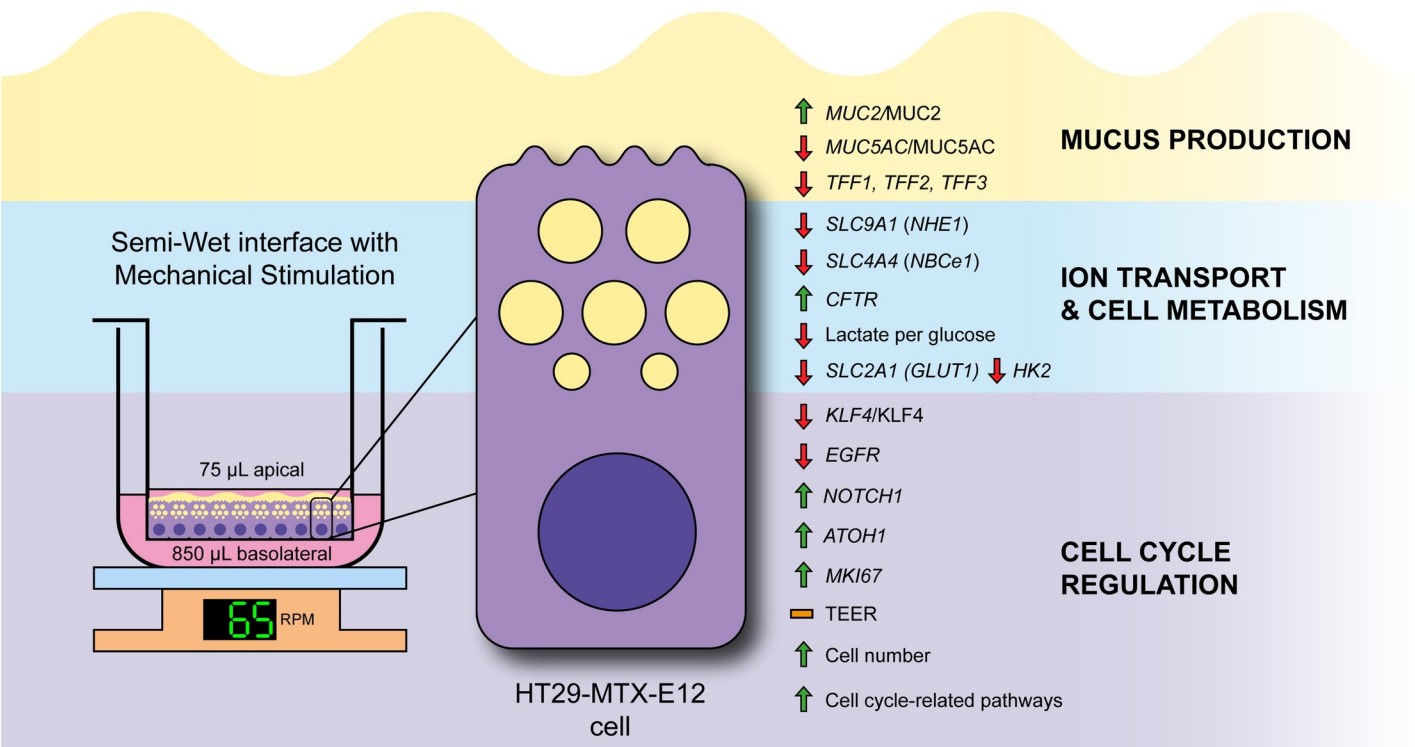

**Fig 8. Graphical summary of effects observed in HT29-MTX-E12 cells grown under SWMS compared to static conditions at t = 15 days.** HT29-MTX-E12 cells were grown on Transwell under static and SWMS conditions. The most important changes between static and SWMS at t = 15 days are depicted. *MUC2*/MUC2 = mucin 2; *MUC5AC*/MUC5AC = mucin 5; *TFF1/2/3* = trefoil factor 1/2/3; *SLC9A1* (*NHE1*) = Na⁺/H⁺ Exchanger 1; *SLC4A4* (*NBCe1*) = Na⁺/HCO3⁻ cotransporter 1; *CFTR* = Cystic Fibrosis transmembrane conductance regulator; lactate:glucose = amount of lactate (mole) produced per mole of glucose consumed in conditioned medium; *SLC2A1* (*GLUT1*) = Glucose Transporter Type 1; *HK2* = Hexokinase 2; *KLF4*/KLF4 = Kruppel like factor 4; *EGFR* = epidermal growth factor receptor; *NOTCH1* = Notch receptor 1; *ATOH1* = atonal BHLH transcription factor *1*; *MKI67* = marker of proliferation Ki-67; TEER = transepithelial electrical resistance.

The transcriptomic analysis of HT29-MTX-E12 cells revealed that SWMS conditions resulted in a strong upregulation of genes and pathways related to cell cycle regulation. In line with this finding, we found higher cell numbers after SWMS culturing. Although the effect size was lower, similar results were found in Caco-2 cells. Similar to our results, a previous study using ALI–which is similar to the semi-wet interface part of SWMS conditions–also showed an increase in cell number and cell layer thickness in intestinal porcine epithelial cells [41], and increased cell proliferation in intestinal organoids [74]. Our experiments can, however, not answer whether the higher cell count is due to an increased height and/or a columnar shape of cells grown under SWMS, as seen in both ALI and SWMS cultures [27, 41] or due to stacking of the cells, although the concentration of cells in the centre of the well point to the latter. In any case, the shared upregulation of cell cycle-related pathways between colon carcinoma cell lines that are so different at the transcriptomic level [75], emphasizes the SWMS-specific effect on cell growth, independent of the cell type. As cell culturing conditions may also strongly influence cell morphology and tight junction formation [76], we explored if the SWMS method resulted in an increased resistance over the epithelial membrane by performing TEER measurements. TEER values of HT29-MTX-E12 cells were comparable to results from earlier studies [26, 77]. No differences were found between SWMS and static conditions for HT29-MTX-E12 cells during the culturing period, which was supported by the low number of differentially regulated genes related to intestinal barrier integrity. These findings are opposing the results from Navabi *et al.*, as they found a slight, but significant, increase in TEER values of HT29-MTX-E12 cells cultured under SWMS conditions. However, apart from the fact that measurements were performed at 21 days, Navabi *et al.* also used a different device (Ussing chamber) for TEER measurements. Furthermore, the higher concentration of cells in the centre of the Transwell as a result of SWMS conditions may have resulted in an underestimation of TEER values, as the chopstick electrodes do not measure the resistance over the whole membrane. In addition to TEER measurements, we also carried out FITC-dextran assays with two probe sizes. The significantly increased paracellular permeability of cells grown under SWMS compared to static conditions is remarkable. Since we do not know whether the mucus layer can interfere with FITC-dextran, we decided to also wash of the mucus layer with the mucolytic agent NAc. This pre-treatment resulted in an unchanged paracellular permeability between both conditions. Nevertheless, the seemingly increased mucus permeability under SWMS, as quantified using FITC-dextran probes, is interesting, but require further investigation with more robust probes to study mucus penetrability, e.g. fluorescent beads [78] that do not chemically interact with mucus.

In parallel to cell proliferation, we also investigated the potential change in cell differentiation pathways as a result of the SWMS method. We focussed on the Notch/Atoh1 pathways, since these are key pathways in the decision between epithelial cell development into either absorptive or secretory cell types [53]. Given the MUC2-promoting effects of SWMS conditions in HT29-MTX-E12 cells, we hypothesized that SWMS conditions inhibited Notch and promoted Atoh1, thereby favouring the differentiation of secretory goblet cells. The interplay between Notch and Atoh in the context of goblet cell differentiation was underscored by the effect of the γ-secretase inhibitor DAPT, which further enhanced mucus production in HT29-MTX-E12 cells by indirectly inhibiting Notch, and thus promoting goblet cell differentiation [27]. Although we found a number of significantly differentially expressed target genes of both Notch and Atoh1, the results did not point towards one particular overrepresented pathway. Based on these results, we suppose that the increased MUC2 production in HT29-MTX-E12 cells was not the result of a change towards a favoured secretory cell fate. This was further supported by a significant downregulation of the goblet cell marker KLF4 at both protein and gene level. The downregulation of *KLF4*/KLF4, identified as a cell cycle

checkpoint protein and negative regulator of cell growth [79, 80], matches with the observed increased cell cycle regulation under SWMS conditions. However, based on our results, we cannot identify cause-effect relations and point at the exact trigger that led to decreased *KLF4*/ KLF4 expression.

Proper mucus production depends on the activity of ion transport [68] and ALI-models with other cell types have shown changes in ion transport [44, 81–83]. Therefore, the significant regulation of several key ion transporters in both HT29-MTX-E12 and Caco-2 cells in our microarray analysis, is interesting. It should be considered, however, that the microarray results are limited to one time point; in this case at which a lower pH is measured under SWMS versus static conditions. Therefore, we cannot conclude whether the change in mucus phenotype can be (partially) explained by differential regulation of ion transporters, or that this regulation is a consequence of the microclimate at the timepoint of analysis. Moreover, our method to measure pH did not allow us to measure potential local pH differences, as seen *in vivo* [84], which could further clarify the observed changes

It has been demonstrated previously that both the semi-wet conditions/ALI and the mechanical stimulation part of SWMS separately, support a more aerobic cell culture environment in non-intestinal cell lines [43–46]. More recent studies confirmed increased oxygen supply and oxidative phosphorylation, concomitant with suppressed glycolysis in intestinal epithelial cells from porcine origin [41, 42]. Similar to the study by Klasvogt *et al.* [42], we measured significantly lower lactate levels and decreased glucose consumption in HT29-MTX-E12 cells grown under SWMS conditions. Interestingly, as opposed to Klasvogt *et al.*, we did not find a downregulation of the HIF-1α gene, but GSEA revealed significant enrichment of the HIF-1 signalling pathway among the downregulated pathways in both HT29-MTX-E12 and Caco-2 cells. This, together with the significantly decreased lactate-per glucose ratio, suggests that under SWMS conditions, cells switch to a more aerobic cell metabolism. Increased mucus production has been observed in other ALI-models (without mechanical stimulation), such as murine gastric surface mucous cells [46] and airway cells from different origins [85].

Mechanical stimulation as part of the SWMS conditions, induced by continuous shaking at 65 rpm, led to a continuous exposure to shear stress. In a model highly similar to SWMS, the shear stress value was calculated to be $1.6 \times 10^{-2} \pm 4.7 \times 10^{-3}$ dyne/cm$^2$ [86]. In other *in vitro* and *ex vivo* models, shear stress values between 1.3 and $2.0 \times 10^{-2}$ dyne/cm$^2$ were sufficient to increase MUC2 protein or gene expression in Caco-2 cells, colon organoids or enteroids compared to static conditions [28, 87]. Additionally, a recent study demonstrated increased mucus thickness of HT29-MTX cells cultured under physiologically relevant shear stress (0.009 dyne/ cm$^2$ [28]), with changes in relative *MUC2* and *MUC5AC* expression compared to static conditions over time [88]. Although we were not able to replicate MUC2-inducing effects in Caco-2 and our model cannot be directly compared to aforementioned models in terms of mechanical stimulation and culture time, we cannot exclude the possibility that shear stress played a role in the observed increase in mucus production under SWMS. Interestingly, next to its role in cell differentiation and cell cycle arrest [89], KLF4 is also known as a mechanosensitive transcription factor in vascular endothelial cells [90], osteoblasts [91] and a dermal cell line [92], in a context-dependent manner. To the best of our knowledge, there is no literature available providing evidence for a link between shear stress and KLF4 in intestinal tissues or models. Moreover, the relation between KLF4 and significantly regulated cytoskeleton-related and mechanosensitive genes (panels listed in S1 File) remains to be further investigated.

[92, 93] An important limitation of our study is that our microarray was restricted to a single time point, whereas our study aimed to capture an overview of the cellular processes affected by the SWMS culture method. To increase insight in gene expression over time (e.g. with regard to cell proliferation or mucus production), future studies should include multiple

time points. Another limitation is that gene expression does not always match with protein expression or activity, though protein levels of MUC2, MUC5AC and KLF4 supported the changes observed at gene levels in both HT29-MTX-E12 and Caco-2 cells. Apart from technical limitations of our study, we demonstrate that HT29-MTX-E12 cells grown under SWMS conditions, as a model developed to better represent the *in vivo* intestine in terms of MUC2 and cell morphology [27] still has its limitations. For instance, the downregulation of genes encoding other mucus-associated proteins, such as TFF3, suggest a less representative mucus layer in terms of whole mucus composition. Additionally, although we focussed primarily on secreted mucins, transmembrane mucins also contribute to a healthy mucosal barrier and prevent the invasion of pathogens [94], amongst a myriad of other functions (reviewed in a.o. [95]. In our transcriptomic analysis, however, *MUC1* was not significantly regulated under SWMS conditions and *MUC4* was only expressed at low levels in HT29-MTX-E12 cells. Though, protein expression of these mucins, as well as their contribution to the model, would be worthwhile studying in future experiments. Still, the function of the SWMS-produced mucus layer as a physical barrier was demonstrated by its interference with the genotoxic activity of colibactin, a toxin produced by certain *Escherichia coli* strains [32]. In a similar fashion, the model could be applied to evaluate the diffusion of drugs and other particles [96]. Furthermore, the model could be useful to study interactions with both pathogenic and commensal bacteria, but would further require quantification of the mucin glycans present, as these have demonstrated to play a crucial role in mucin-microbe interactions [97].

Altogether, we confirm the usefulness of SWMS cell culture conditions to improve the *in vivo* representation of the mucus layer *in vitro*, with regard to the increase in intestinal mucin MUC2. Our study provides insight in potential underlying processes, which might ultimately lead to a step-by-step improvement of the representativeness of the *in vitro* mucus layer. For instance, our study demonstrates upregulation of cell cycle regulation, downregulation of KLF4, differential regulation of ion transporters and increased aerobic metabolism of cells cultured under SWMS versus static conditions. Further research should also focus on the qualitative aspects of the *in vitro* mucus layer, for example with regard to mucin glycosylation, disulphide bonds that assure firmness of the mucus, and the contribution of other proteins present in the mucus layer, such as TFFs, as observed *in vivo*.

## Supporting information

**S1 Fig. Overview microarray results in HT29-MTX-E12, Caco-2 and LS174T cells grown under static and/or SWMS conditions. A)** Volcano plot highlighting the Log Fold Change (logFC) on the x-axis and the corresponding *p*-values (-log(10)) on the y-axis for the comparison SWMS versus static conditions in HT29-MTX-E12 cells and **B)** Caco-2 cells. *n* = 3. (PDF)

**S2 Fig.** Overview of Dot Blot results for A) MUC2 and B) MUC5AC in HT29-MTX-E12 cells cultured under static and SWMS conditions and LS174T cells. A concentration of 68.2 μg/mL was blotted and seven times serially diluted 1:2. In column 4, 8 and 12, PBS was used as a negative control. C) Images of Ponceau Red staining (colorimetric and photographic) that were used as reference for total protein content. Protein density was based on the colorimetric image. D) Protein expression of MUC2 and MUC5AC in LS174T cells, expressed as density (a. u.) per ug/mL protein blotted, after correction of Ponceau Red density (*n* = 3) E) Ratio of MUC2 and MUC5AC protein expression in LS174T cells (*n* = 3). Original Dot Blots for Caco-2 are displayed in F) for MUC2 and G) for MUC5AC. Caco-2 data are indicated with a blue box. HT29-MTX-E12 and LS174T samples have been repeated including additional replicates

for figure A and B.
(PDF)

**S3 Fig. Venn diagram showing the number of shared and unique differentially expressed up- and downregulated genes (-1.5 $\geq$ Fold change $\geq$ 1.5) between HT29-MTX-E12 and Caco-2 cells cultured under static and SWMS conditions.** The top 20 up- and downregulated genes shared between HT29-MTX-E12 and Caco-2 cells is given in the tables.
(PDF)

**S4 Fig. Overview of cell-cycle regulation parameters in HT29-MTX-E12 and Caco-2 cells under static and SWMS conditions. A)** Microarray gene expression values (Log2) of *MKI67* in HT29-MTX-E12 and Caco-2 cells cultured under static and SWMS conditions. **B)** Cell count after t = 15 days, expressed as cells per $cm^2$, of HT29-MTX-E12 and Caco-2 cells cultured under static and SWMS conditions. **** $p < 0.0001$.
(PDF)

**S5 Fig. Western immunoblotting results of KLF4, including all three biological replicates (batch 1, 2 and 3) in Caco-2, HT29-MTX-E12 and LS174T-cells grown under static and SWMS conditions or static only (LS174T).** HSP90 was used as house-keeping protein.
(PDF)

**S6 Fig. Images of HT29-MTX-E12 and Caco-2 cells grown under static and SWMS conditions.** A) Pictures of HT29-MTX-E12 and Caco-2 cells grown under static and SWMS conditions at t = 15 days. B) Bright-field microscopy pictures (20x) of HT29-MTX-E12 and Caco-2 cells grown under static and SWMS condition, focussed on the centre of the Transwell membranes.
(PDF)

**S7 Fig. pH data and calibration of HT29-MTX-E12 and Caco-2 cells under static and SWMS conditions. A)** Ratio of Absorbance of cell culture medium (415 and 560 nm) of HT29-MTX-E12 and Caco-2 cells under static and SWMS conditions measured at 5% $CO_2$. **B)** Medium pH of apical and basolateral compartments of HT29-MTX-E12 cells grown under static and SWMS conditions at t = 1–14 days. * $p < 0.05$; ** $p < 0.01$, $n = 3$ **C)** Standard curve of pH values and absorbance values of cell culture medium measured at 415/560 nm at 5% $CO_2$. **D)** The linear part the standard curve, including trendline.
(PDF)

**S8 Fig. Lactate and glucose concentrations of HT29-MTX-E12 cells grown under static and SWMS conditions. A)** Lactate concentration (mM) in cell culture medium of HT29-MTX-E12 cells collected during every medium refreshing moment. **B)** Total lactate (micromole) produced per well in medium collected from apical and basolateral compartments of HT29-MTX-E12 grown under static or SWMS conditions, at t = 1–14 days. **C)** Total glucose (micromole) consumed per well from medium collected from apical and basolateral compartments of HT29-MTX-E12 grown under static and SWMS conditions, at t = 1–14 days.
(PDF)

**S1 File. Panels of genes, including mucins, goblet cell markers, epithelial barrier-related genes, KLF4-target genes, ion transporters, cytoskeleton and mechanosensitive genes in HT29-MTX-E21 cells.** The RMA values of the three independent biological replicates are given, together with the corresponding fold changes and p-values between SWMS and static conditions.
(XLSX)

**S2 File. Up- and downregulated pathways between SWMS and static conditions in HT29-MTX-E12 and Caco-2 cells, obtained by Gene Set Enrichment Analysis (GSEA).** NES = Normalized Enrichment Score; FDR = False Discovery Rate. (XLSX)

**S1 Raw images.** (PDF)

## Author Contributions

**Conceptualization:** Janneke Elzinga, Benthe van der Lugt, Clara Belzer, Wilma T. Steegenga.

**Funding acquisition:** Wilma T. Steegenga.

**Investigation:** Janneke Elzinga, Benthe van der Lugt.

**Methodology:** Janneke Elzinga, Benthe van der Lugt.

**Supervision:** Clara Belzer, Wilma T. Steegenga.

**Validation:** Janneke Elzinga, Benthe van der Lugt.

**Visualization:** Janneke Elzinga, Benthe van der Lugt.

**Writing – original draft:** Janneke Elzinga, Benthe van der Lugt.

**Writing – review & editing:** Janneke Elzinga, Benthe van der Lugt, Clara Belzer, Wilma T. Steegenga.

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
