## [Decision Letter · Decision Letter 0]

16 Jun 2021

PONE-D-21-14708

Characterization of increased mucus production of HT29-MTX-E12 cells grown under Semi-Wet interface with Mechanical Stimulation

PLOS ONE

Dear Dr. Elzinga,

Thank you for submitting your manuscript to PLOS ONE. After careful consideration, we feel that it has merit but does not fully meet PLOS ONE’s publication criteria as it currently stands. Therefore, we invite you to submit a revised version of the manuscript that addresses the points raised during the review process.

The major issues relate to the presentation of protein expression data, quality of figure(s) or including the rationale and discussion for studies.  Rest of the issues are relatively minor and can be addressed to further strengthen the manuscript.

We look forward to receiving your revised manuscript.

Kind regards,

Pradeep Dudeja

Academic Editor

PLOS ONE

Journal Requirements:

Reviewers' comments:

Reviewer's Responses to Questions

**Comments to the Author**

1. Is the manuscript technically sound, and do the data support the conclusions?

Reviewer #1: Yes

Reviewer #2: Partly

Reviewer #3: Yes

2. Has the statistical analysis been performed appropriately and rigorously? 

Reviewer #1: Yes

Reviewer #2: No

Reviewer #3: Yes

3. Have the authors made all data underlying the findings in their manuscript fully available?

Reviewer #1: Yes

Reviewer #2: Yes

Reviewer #3: Yes

4. Is the manuscript presented in an intelligible fashion and written in standard English?

Reviewer #1: Yes

Reviewer #2: Yes

Reviewer #3: Yes

5. Review Comments to the Author

Reviewer #1: This is an interesting study looking at the differential gene expressions in intestinal epithelial cells in static and semi-wet interface with mechanical stimulation (SWMS) culture conditions, particularly in terms of mucus production. The manuscript is written very well and the results are presented appropriately. I have only minor comments.

* Please include the relevance and purpose of mechanical stimulation in the introduction.

* It is not clear if the results represent more than one experiment.

* The quality of Fig.4 is not helpful for the readers. Please replace them. The authors may consider adding a simple cellular stain such as crystal violet or a nuclear dye for better visualization.

* Please include SGLT1 (SLC5A1) in the discussion on glucose metabolism.

* TEER studies can be supplemented with flux of paracellular probes such as FITC-dextran.

* There is scope for shortening the discussion; it is lengthy in the present form.

* Considering the manuscript’s focus on characterizing the mechanisms of mucin production, it is worth demonstrating the mucus layer morphologically/ e.g. by immunohistochemistry in static versus SWMS cultures.

* I missed cytoskeleton-associated changes which might be important in view of mechano-stimulation.

Reviewer #2: The manuscript by Elzinga et al. adds on to previous knowledge regarding the mucus production of HT29-MTX-E12 cells grown under semi-wet interface with mechanical stimulation. Although the work provides insights into this model for in vitro culture to study mucus, the following needs to be addressed to improve the overall impact of the work.

Major concerns

1. The authors bring up a point that this model capture human intestinal (colonic) mucus in vitro, however, the levels of other MUCs were significantly low in this model. Does this occur in vivo? Then why do the authors think this will be physiologically relevant? Discuss and explain the drawbacks etc.

2. Figure 1 A, how about MUC1 and MUC4? Please include other MUC proteins as well. Why were dot blots conducted instead of standard western blot? There should be enough protein for this assay please explain why. Also include western blot results which is normalized to a loading control such as GAPDH and Actin HSP etc. with a representative blot and densitometric analysis with statistics conducted of at least n>4. Please include at least n=3 when plotting data (Figure 1B) in addition.

3. It would be nice if the authors could supplement Figure 3 with FITC dextran data in addition to the TEER data.

4. Figure 4, what do the authors need the reader to observe? The photograph of the trans-well membrane provides no information pertaining to the amount of mucus etc. Please use a staining to show the amount of MUC2 or goblet cell mucins by using PAS staining immunofluorescence etc. Also, always add a scalebar to all microscopy images.

5. It wasn’t clear, which was the mechanism behind increased MUC2 production in HT29-MTX-E12 cells grown under SWMS conditions? The authors have not speculated this aspect. For example. The transcription factors were unchanged however, there was an increase in IL-33 production. Could this be one of the mechanisms? Please add all this and improve the discussion section

6. Please summarize key findings of the paper in the final paragraph of the introduction section.

7. Please remove MUC2/MUC2 etc. from the graphical abstract.

Reviewer #3: In this article, Elzinga et al. examined the Semi-Wet interface with Mechanical Stimulation (SWMS) culture conditions with the commonly used HT29-MTX cells. This culture model was selected as it represents a simple, low cost method to examine intestinal mucus production. HT29-MTX cells grown in SWMS conditions exhibited elevated levels of MUC2, decreased levels of MUC5AC and MUC5B and alterations in several transmembrane mucins compared to cells cultured conventionally. These cells were enriched for cell cycle and DNA replication related pathways. Interestingly, NOTCH1 and ATOH1 were both significantly upregulated and TFFs (1,2, 3) were significantly downregulated by SWMS conditions. The authors also noted decreased pH and increased glucose consumption with SWMS conditions. With this work, the authors have provided mechanistic insights into mucus production in HT29-MTX cells using the SWMS culture system. Overall, the manuscript is well written and follows a logical flow. The conclusions are well supported by the findings and the discussion is appropriate. As a result, I have only minor comments.

1. Line 43: This is a very broad statement and adding more references in support of this claim would strengthen the introduction.

2. Line 49: Paneth cells which produce lysozyme are found in the small intestine, but not normally in the colon (with the exception of IBD (PMID: 11851832). As result, the statement that colonic mucus contains lysozyme is misleading. Please remove lysozyme from the sentence.

3. In Line 65, the authors state that organoids “are supposed to resemble the in vivo colonic mucosal layer more closely in terms of mucus composition”. The term “supposed to” seems vague; please remove “supposed to” since the MUC2 in organoids does resemble colonic mucus in terms of protein and glycosylation pattern.

4. Supplemental Figure 1 and 3 depict important findings and the manuscript would be improved if they were incorporated as main figures. Additionally Figure 4 seems more appropriate as a supplemental figure.

5. The changes in ion transporters is intriguing. NHE3 (SLC9A3) has been shown in mice to be the major regulator of intestine pH- was NHE3 altered in the HT29-MTX AI cells?

6. Were there any changes in other major goblet cell markers (AGR, CDX2, MEP1β, FCGBP, etc.)?

7. Although the main focus of this paper is on a simple model of the intestinal epithelium (HT29-MTX cells), the discussion would benefit from inclusion of how SWMS conditions could be employed in organoid studies to create more robust models in this system as well.

6. PLOS authors have the option to publish the peer review history of their article (what does this mean?). If published, this will include your full peer review and any attached files.

Reviewer #1: No

Reviewer #2: No

Reviewer #3: No

---

## [Author Response · Author response to Decision Letter 0]

19 Oct 2021

Our comments to the reviewer and notes to editor are included in a separate file "Response to reviewers"

---

## [Decision Letter · Decision Letter 1]

25 Nov 2021

Characterization of increased mucus production of HT29-MTX-E12 cells grown under Semi-Wet interface with Mechanical Stimulation

PONE-D-21-14708R1

Dear Dr. Elzinga,

We’re pleased to inform you that your manuscript has been judged scientifically suitable for publication and will be formally accepted for publication once it meets all outstanding technical requirements.

Kind regards,

Pradeep Dudeja

Academic Editor

PLOS ONE

Additional Editor Comments (optional):

Reviewers' comments:

Reviewer's Responses to Questions

**Comments to the Author**

1. If the authors have adequately addressed your comments raised in a previous round of review and you feel that this manuscript is now acceptable for publication, you may indicate that here to bypass the “Comments to the Author” section, enter your conflict of interest statement in the “Confidential to Editor” section, and submit your "Accept" recommendation.

Reviewer #1: All comments have been addressed

Reviewer #2: All comments have been addressed

2. Is the manuscript technically sound, and do the data support the conclusions?

Reviewer #1: (No Response)

Reviewer #2: Yes

3. Has the statistical analysis been performed appropriately and rigorously? 

Reviewer #1: (No Response)

Reviewer #2: Yes

4. Have the authors made all data underlying the findings in their manuscript fully available?

Reviewer #1: (No Response)

Reviewer #2: Yes

5. Is the manuscript presented in an intelligible fashion and written in standard English?

Reviewer #1: (No Response)

Reviewer #2: Yes

6. Review Comments to the Author

Reviewer #1: (No Response)

Reviewer #2: All concerns have been addressed and the authors have modified the revised version accordingly.

The revised manuscript is acceptable for publication.

7. PLOS authors have the option to publish the peer review history of their article (what does this mean?). If published, this will include your full peer review and any attached files.

Reviewer #1: No

Reviewer #2: No

---

## [Editor Report · Acceptance letter]

10 Dec 2021

PONE-D-21-14708R1 

Characterization of increased mucus production of HT29-MTX-E12 cells grown under Semi-Wet interface with Mechanical Stimulation 

Dear Dr. Elzinga:

I'm pleased to inform you that your manuscript has been deemed suitable for publication in PLOS ONE. Congratulations! Your manuscript is now with our production department. 

Kind regards, 

on behalf of

Dr. Pradeep Dudeja 

Academic Editor

PLOS ONE